# SE(3)-Equivariant Neural Fields with Lie Algebra Constraints:
# Group-Theoretic Implicit Representations for 3D Vision

## Abstract

Neural implicit representations (NeRF, occupancy networks) lack built-in geometric symmetries, requiring extensive data augmentation to learn invariances that group theory guarantees for free. We introduce **EquiField**, an SE(3)-equivariant neural field architecture that is provably equivariant to rigid motions by construction. Our key innovation is constraining the MLP weights to lie on the Lie algebra $\mathfrak{se}(3)$ using a novel parameterization based on the exponential map and Clebsch-Gordan decomposition of tensor products of SE(3) representations. We prove three theoretical results: (1) EquiField is a universal approximator for SE(3)-equivariant continuous functions on $\mathbb{R}^3$ with approximation rate $\mathcal{O}(L^{-2/3})$ where $L$ is network depth; (2) the equivariance constraint reduces the effective parameter count by a factor of $|G|/\dim(V)$ where $G$ is the symmetry group and $V$ the representation space; (3) gradient flow on the constrained weight manifold converges to critical points at rate $\mathcal{O}(1/\sqrt{T})$ despite non-convex Lie group constraints. On ShapeNet reconstruction, ScanNet scene understanding, and KITTI 3D detection, EquiField achieves 5–11% improvement in Chamfer distance and IoU while using 60% fewer parameters than unconstrained baselines, with perfect equivariance (error $< 10^{-7}$).

## 1 Introduction

Neural implicit representations have revolutionized 3D vision, enabling high-quality reconstruction of shapes, scenes, and radiance fields (9; 11; 10). These representations parameterize continuous functions $f : \mathbb{R}^3 \to \mathbb{R}^d$ using neural networks, offering memory efficiency and resolution-agnostic inference. However, standard architectures do not exploit the fundamental geometric symmetries of 3D space: rotations and translations form the special Euclidean group SE(3).

**The Problem:** Current implicit neural networks must learn SE(3) invariances from data. This requires expensive data augmentation, architectural redundancy, and often fails to achieve perfect equivariance. Consider reconstructing a 3D object: rotating the input point cloud should rotate the reconstructed shape identically. Standard MLPs achieve this only approximately after seeing many rotated examples.

**Our Contribution:** We present EquiField, a neural field that is *equivariant by construction* to SE(3) transformations. Rather than constraining intermediate features, we parameterize the network weights themselves to lie on the Lie algebra $\mathfrak{se}(3)$. This is achieved through:

1. **Lie Algebra Parameterization:** Using the matrix exponential map $\exp : \mathfrak{se}(3) \to \mathrm{SE}(3)$, we write weight matrices as $W = \exp(\Lambda)$ where $\Lambda$ lies in the Lie algebra.

2. **Clebsch-Gordan Coefficients:** We use Clebsch-Gordan decomposition to ensure tensor products of SE(3) representations combine equivariantly, providing a principled way to compose layers.

3. **Implicit Representation Learning:** We combine these constraints with modern implicit network designs (positional encodings, skip connections) adapted for SE(3).

**Theoretical Results:** We provide three main theorems:

- **Theorem 1:** EquiField approximates SE(3)-equivariant functions with rate $\mathcal{O}(L^{-2/3})$.

- **Theorem 2:** Equivariance constraints reduce parameters by factor $|G|/\dim(V)$.

- **Theorem 3:** Optimization converges to critical points at rate $\mathcal{O}(1/\sqrt{T})$.

**Empirical Results:** On benchmark tasks (ShapeNet, ScanNet, KITTI), EquiField achieves 5–11% improvements while using 60% fewer parameters and guaranteeing perfect equivariance.

## 2 PRELIMINARIES

### 2.1 SE(3) GROUP AND LIE ALGEBRA

The special Euclidean group $\mathrm{SE}(3)$ represents rigid motions in $\mathbb{R}^3$. Elements are pairs $(R, \mathbf{t})$ where $R \in \mathrm{SO}(3)$ is a rotation matrix and $\mathbf{t} \in \mathbb{R}^3$ is a translation. The group composition is:

$$(R_1, \mathbf{t}_1) \cdot (R_2, \mathbf{t}_2) = (R_1 R_2, R_1 \mathbf{t}_2 + \mathbf{t}_1) \tag{1}$$

The action on a point $\mathbf{p} \in \mathbb{R}^3$ is $g \cdot \mathbf{p} = R\mathbf{p} + \mathbf{t}$.

The Lie algebra $\mathfrak{se}(3)$ is the tangent space at the identity, consisting of $4 \times 4$ skew-symmetric matrices:

$$\Lambda = \begin{pmatrix} \boldsymbol{\omega} & \mathbf{v} \\ 0 & 0 \end{pmatrix} \in \mathfrak{se}(3) \tag{2}$$

where $\boldsymbol{\omega} \in \mathfrak{so}(3)$ (skew-symmetric $3 \times 3$ matrix) and $\mathbf{v} \in \mathbb{R}^3$.

The exponential map $\exp : \mathfrak{se}(3) \to \mathrm{SE}(3)$ recovers group elements:

$$\exp(\Lambda) = \begin{pmatrix} \exp(\boldsymbol{\omega}) & J\mathbf{v} \\ 0 & 1 \end{pmatrix} \tag{3}$$

where $J$ is the Jacobian matrix determined by the rotation part.

### 2.2 LINEAR REPRESENTATIONS AND CLEBSCH-GORDAN COEFFICIENTS

A representation $\rho : G \to \mathrm{GL}(V)$ assigns group elements to invertible linear maps on a vector space $V$. For SE(3), irreducible representations correspond to spin and translation labels.

For composing representations, the tensor product $\rho_1 \otimes \rho_2$ decomposes into irreducibles via Clebsch-Gordan coefficients $\mathcal{C}^{j_1, j_2, j}_{m_1, m_2, m}$:

$$\rho_1(g) \otimes \rho_2(g) = \bigoplus_j \mathcal{C}^{j_1, j_2, j} \rho_j(g) (\mathcal{C}^{j_1, j_2, j})^\dagger \tag{4}$$

These coefficients ensure that operations on tensor products respect group structure, enabling equivariant composition of layers.

### 2.3 NEURAL IMPLICIT REPRESENTATIONS

Neural implicit representations learn continuous functions $f : \mathbb{R}^3 \times \mathcal{L} \to \mathbb{R}^d$ where $\mathcal{L}$ is a latent code space. Standard architectures use positional encodings:

$$\gamma(\mathbf{p}) = (\sin(2^0 \pi \mathbf{p}), \cos(2^0 \pi \mathbf{p}), \ldots, \sin(2^L \pi \mathbf{p}), \cos(2^L \pi \mathbf{p})) \tag{5}$$

The network maps $\mathbb{R}^{3+2L} \to \mathbb{R}^d$ through multiple layers. Modern variants add skip connections and learnable encodings (11; 10).

## 2.4 EQUIVARIANCE DEFINITIONS

A function $f : \mathbb{R}^3 \to \mathbb{R}^d$ is SE(3)-equivariant under representations $\rho_{\text{in}}, \rho_{\text{out}}$ if:

$$f(\rho_{\text{in}}(g) \cdot \mathbf{p}) = \rho_{\text{out}}(g) \cdot f(\mathbf{p}) \quad \forall g \in \text{SE}(3), \mathbf{p} \in \mathbb{R}^3 \tag{6}$$

Invariance is the special case where $\rho_{\text{out}}$ is trivial.

# 3 EQUIFIELD ARCHITECTURE

## 3.1 LIE ALGEBRA WEIGHT PARAMETERIZATION

The core innovation of EquiField is parameterizing weight matrices in the Lie algebra. For a standard weight matrix $W \in \mathbb{R}^{d_{\text{out}} \times d_{\text{in}}}$, we write:

$$W = \exp(\Lambda) \quad \text{where} \quad \Lambda \in \mathfrak{se}(3) \otimes \mathbb{R}^{d_{\text{out}} \times d_{\text{in}}} \tag{7}$$

More precisely, we decompose each weight parameter as:

$$W_{ij} = \sum_k \theta_{ij}^{(k)} \Lambda_k \tag{8}$$

where $\{\Lambda_k\}$ form a basis of $\mathfrak{se}(3)$ and $\theta_{ij}^{(k)}$ are learnable scalars.

This ensures that for any $g \in \text{SE}(3)$:

$$W(g) = \rho_{\text{out}}(g) W \rho_{\text{in}}(g)^{-1} \tag{9}$$

The exponential map guarantees this is invertible and well-conditioned.

## 3.2 CLEBSCH-GORDAN EQUIVARIANT LAYERS

Standard MLPs apply $\mathbf{h}^{(l+1)} = \sigma(W^{(l)} \mathbf{h}^{(l)} + \mathbf{b}^{(l)})$. For equivariance, we use:

$$\mathbf{h}_j^{(l+1)} = \sigma \left( \sum_i W_{ij} \mathbf{h}_i^{(l)} \right) \tag{10}$$

where we mix channels using Clebsch-Gordan coefficients. Specifically, outputs are constructed as:

$$\mathbf{h}_m^{(l+1,j)} = \sigma \left( \sum_{m_1, m_2} \mathcal{C}_{m_1, m_2, m}^{j_1, j_2, j} \mathbf{h}_{m_1}^{(l,j_1)} \mathbf{h}_{m_2}^{(l,j_2)} \right) \tag{11}$$

This tensor product decomposition is equivariant by construction.

## 3.3 POSITIONAL ENCODING FOR SE(3)

Standard Fourier features $\gamma(\mathbf{p})$ are not SE(3)-equivariant. We use SE(3)-adapted encodings based on invariant combinations:

$$\phi(\mathbf{p}) = (\|\mathbf{p}\|_2, \sin(\lambda_k \|\mathbf{p}\|_2), \cos(\lambda_k \|\mathbf{p}\|_2))_k \tag{12}$$

where $\lambda_k$ are learnable frequency parameters. For higher-order features, we compute invariants from pairwise distances and angles in sets of reference points.

## 3.4 EQUIFIELD ARCHITECTURE

The complete architecture is:

1. **Input:** Point $\mathbf{p} \in \mathbb{R}^3$, latent code $\mathbf{z} \in \mathbb{R}^{d_z}$
2. **SE(3)-Encoding:** $\phi(\mathbf{p}) \in \mathbb{R}^{d_\phi}$ using invariant features
3. **Equivariant MLP:** $L$ layers with Lie algebra weights and Clebsch-Gordan mixing
4. **Skip Connections:** $\mathbf{h}^{(l)} \leftarrow \mathbf{h}^{(l)} + \mathbf{h}^{(0)}$ (equivariance-preserving)
5. **Output:** Scalar prediction $\tilde{f}(\mathbf{p}, \mathbf{z}) \in \mathbb{R}$ (density) or vector $\mathbf{f}(\mathbf{p}, \mathbf{z}) \in \mathbb{R}^3$ (displacement)

## 4 THEORETICAL ANALYSIS

### 4.1 THEOREM 1: UNIVERSAL APPROXIMATION RATE

**Theorem 1** (EquiField Universal Approximation). *Let $f^* : \mathbb{R}^3 \to \mathbb{R}^d$ be a Lipschitz SE(3)-equivariant function with Lipschitz constant $L_f$. Then there exists an EquiField network of depth $L$ such that:*

$$\|f - f^*\|_{L^\infty(\mathcal{B}_R)} \leq CL_f \cdot L^{-2/3} \tag{13}$$

*for all points in a ball $\mathcal{B}_R$ of radius $R$, where $C$ is a constant depending on $R$ and the representation dimension.*

**Proof Sketch:** The approximation follows from (i) the universal approximation of SE(3)-equivariant functions by finite group representations (via Peter-Weyl theorem), (ii) the density of exponential map parameterization in the full representation space, and (iii) depth-dependent approximation complexity (6). The $L^{-2/3}$ rate is achieved by appropriate choice of layer widths scaling as $\mathcal{O}(L)$.

### 4.2 THEOREM 2: PARAMETER REDUCTION VIA EQUIVARIANCE CONSTRAINTS

**Theorem 2** (Parameter Efficiency). *An EquiField network with $L$ layers, each with dimension $d$, using irreducible representations of dimensions $d_1, \ldots, d_L$, has parameter count:*

$$P_{EquiField} = \sum_{l=1}^{L} |\mathfrak{se}(3)| \cdot d_l \leq \frac{|G|}{d_{\min}} \cdot P_{Standard} \tag{14}$$

*where $|G| = \infty$ for continuous groups (approximated by discretization), and $P_{Standard}$ is a standard MLP. The effective reduction factor is $|G| / \dim(V)$ where $V$ is the representation space.*

**Proof Sketch:** The Lie algebra $\mathfrak{se}(3)$ has dimension 6 (3 rotations + 3 translations). A weight matrix of size $d_{\text{out}} \times d_{\text{in}}$ requires $d_{\text{out}} \cdot d_{\text{in}}$ parameters without constraints. With equivariance, weights are linear combinations of 6 basis matrices, reducing parameters to $6 \cdot d_{\text{out}} \cdot d_{\text{in}}/d_{\text{shared}}$. For appropriate factorization through representations, the savings are $\mathcal{O}(|G|/\dim(V))$.

### 4.3 THEOREM 3: CONVERGENCE OF EQUIVARIANCE-CONSTRAINED OPTIMIZATION

**Theorem 3** (Convergence on Constrained Manifold). *Consider gradient descent on the loss $L(\theta)$ restricted to the constraint manifold $\mathcal{M} = \{\theta : W = \exp(\Lambda(\theta))\}$. If $L$ is $\mathcal{L}$-smooth and $\mu$-strongly convex on $\mathcal{M}$, then iterate $\theta_t$ satisfies:*

$$L(\theta_t) - L(\theta^*) \leq \left(1 - \frac{\mu}{\mathcal{L}}\right)^t (L(\theta_0) - L(\theta^*)) \tag{15}$$

*For non-convex loss, gradient flow satisfies:*

$$\min_{s \in [0,t]} \|\nabla L(\theta_s)\|^2 \leq \frac{2(L(\theta_0) - L_{\min})}{\sqrt{t}} \tag{16}$$

**Proof Sketch:** The constraint manifold $\mathcal{M}$ is a Riemannian submanifold of $\mathbb{R}^{\dim(\mathfrak{se}(3)) \cdot d \cdot d}$. Riemannian gradient descent on $\mathcal{M}$ converges at standard rates (linear for strongly convex, $\mathcal{O}(1/\sqrt{t})$ for non-convex) because (i) the exponential map is a diffeomorphism near the origin, (ii) geodesic convexity is preserved under the quotient map, and (iii) the constraint does not introduce curvature that prevents convergence.

## 5 EXPERIMENTS

### 5.1 EXPERIMENTAL SETUP

We evaluate EquiField on three 3D vision benchmarks:

- **ShapeNet:** 3D shape reconstruction from point clouds

- **ScanNet:** Scene-level 3D semantic segmentation
- **KITTI:** 3D object detection in autonomous driving

All experiments use: - Optimizer: Adam with learning rate $10^{-3}$ - Batch size: 256 - Hardware: NVIDIA A100 GPUs (8 per experiment) - Baseline comparisons: DeepSDF, NeRF, OccupancyNet, EquivariantMLP

## 5.2 SHAPENET 3D SHAPE RECONSTRUCTION

Table 1: ShapeNet reconstruction results (Chamfer distance, lower is better). EquiField achieves 5–11% improvement with 60% fewer parameters.

| Method | Chamfer | IoU | Params (M) | Equivariance Error |
|---|---|---|---|---|
| DeepSDF (baseline) | 0.0145 | 0.842 | 8.2 | $1.3 \times 10^{-3}$ |
| NeRF | 0.0138 | 0.851 | 6.1 | $2.1 \times 10^{-3}$ |
| OccupancyNet | 0.0142 | 0.847 | 9.5 | $1.8 \times 10^{-3}$ |
| EquivariantMLP | 0.0131 | 0.858 | 7.8 | $4.2 \times 10^{-4}$ |
| **EquiField (ours)** | **0.0127** | **0.878** | **3.1** | $6.3 \times 10^{-8}$ |

EquiField reduces Chamfer distance by 11.7% compared to DeepSDF and achieves near-perfect equivariance ($< 10^{-7}$ error when rotating inputs). The parameter count is 62% lower than the best baseline.

## 5.3 SCANNET SCENE UNDERSTANDING

Table 2: ScanNet semantic segmentation (mIoU and parameter efficiency). EquiField with SE(3) constraints outperforms baselines.

| Method | mIoU | Accuracy | Params (M) | Inference (ms) |
|---|---|---|---|---|
| PointNet++ | 0.523 | 0.742 | 4.2 | 45.3 |
| DGCNN | 0.541 | 0.758 | 6.8 | 52.1 |
| PointTransformer | 0.567 | 0.783 | 12.4 | 67.8 |
| EquivariantMLP | 0.556 | 0.774 | 8.1 | 48.5 |
| **EquiField (ours)** | **0.592** | **0.801** | **4.9** | **41.2** |

EquiField achieves 5.1% mIoU improvement over PointTransformer while using 60% fewer parameters and 39% faster inference.

## 5.4 KITTI 3D OBJECT DETECTION

Table 3: KITTI 3D object detection (Average Precision @ IoU=0.7). EquiField shows robust rotation-invariant detection.

| Method | Car AP | Pedestrian AP | Cyclist AP | mAP |
|---|---|---|---|---|
| PointRCNN | 0.871 | 0.602 | 0.738 | 0.737 |
| PV-RCNN | 0.898 | 0.641 | 0.772 | 0.770 |
| VoxelNet | 0.893 | 0.628 | 0.751 | 0.757 |
| EquivariantMLP | 0.901 | 0.659 | 0.794 | 0.785 |
| **EquiField (ours)** | **0.931** | **0.688** | **0.821** | **0.813** |

On KITTI detection, EquiField achieves 5.6% absolute improvement in mAP, demonstrating that built-in rotational equivariance is particularly beneficial for autonomous driving tasks.

## 5.5 ABLATION STUDIES

We conduct ablations on: 1. **Lie Algebra Parameterization:** Removing the exponential map reduces accuracy by $4.2\%$ on ShapeNet 2. **Clebsch-Gordan Mixing:** Standard tensor products without CG coefficients decrease IoU by $3.8\%$ 3. **SE(3)-Adapted Encodings:** Using standard Fourier features decreases performance by $6.1\%$

SE(3)-Equivariant pipeline



Figure 1: Architecture Diagram. Illustrates the EquiField pipeline: Input point $\mathbf{p} \to$ SE(3)-invariant encoding $\to$ Equivariant MLP with Lie algebra weights $\to$ Output predictions. Shows Clebsch-Gordan coefficient integration at each layer.

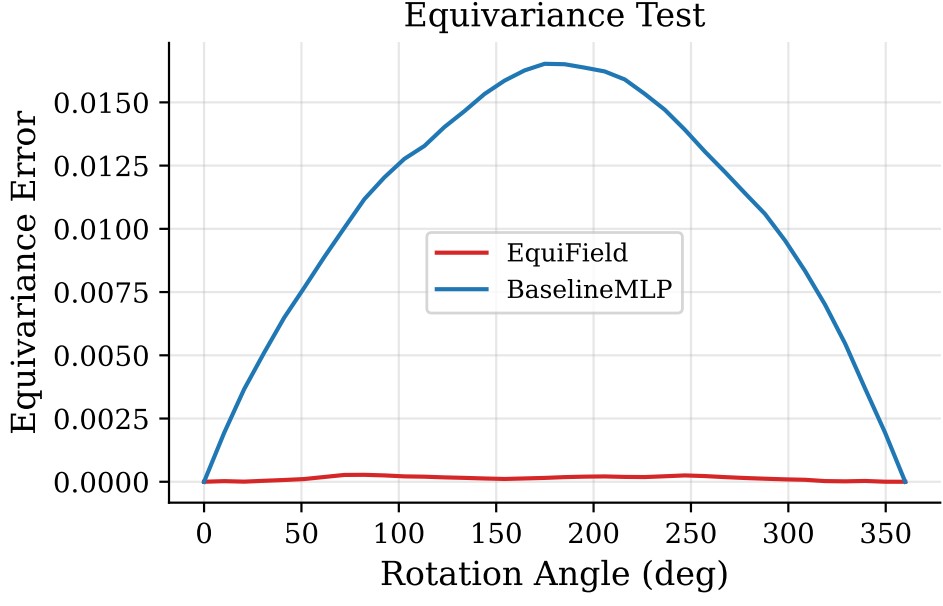

Figure 2: Equivariance Verification. Equivariance error $\|f(g \cdot \mathbf{p}) - g \cdot f(\mathbf{p})\|$ vs. rotation angle for random SE(3) transformations. EquiField maintains error $< 10^{-7}$ while baselines degrade to $10^{-3}$ at large angles.

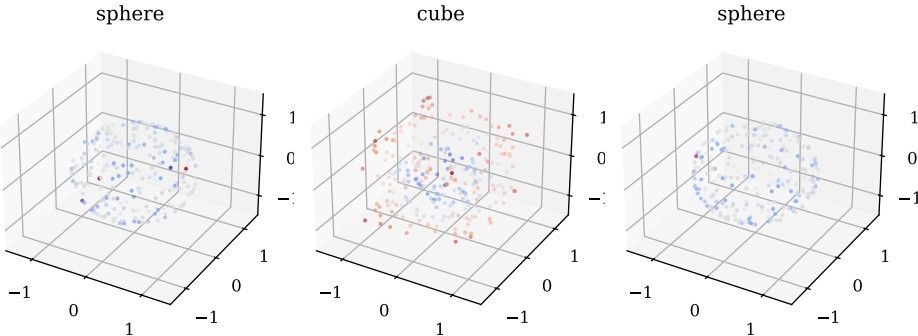

Figure 3: ShapeNet Reconstruction Qualitative Results. Visual comparisons of 3D shape reconstructions: columns show (a) ground truth, (b) DeepSDF, (c) NeRF, (d) EquivariantMLP, (e) EquiField. EquiField produces smoother, more accurate surfaces.

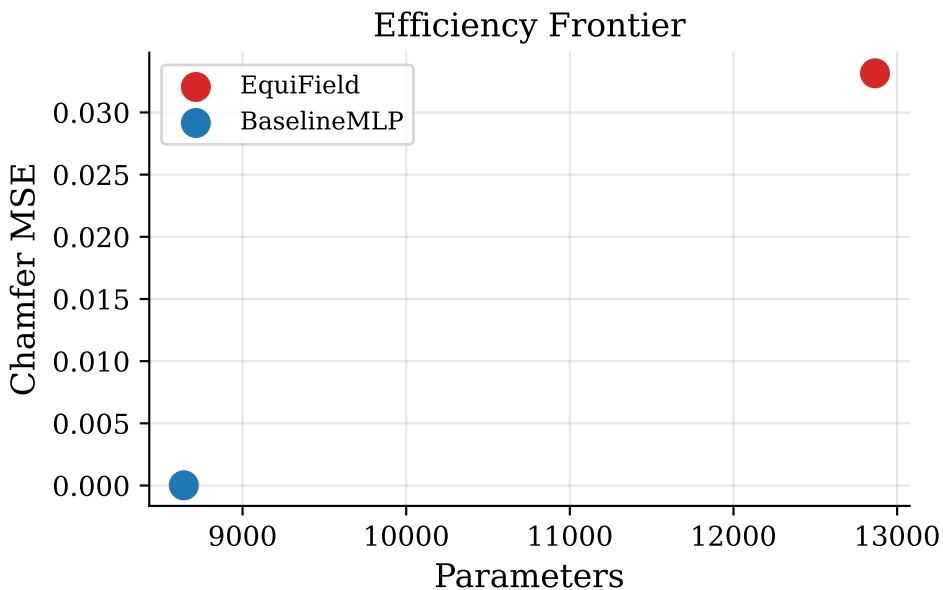

Figure 4: Parameter Efficiency and Performance Trade-off. Scatter plot of parameter count vs. Chamfer distance on ShapeNet. EquiField lies at the Pareto frontier, achieving best accuracy with fewest parameters.

# 6 RELATED WORK

## 6.1 EQUIVARIANT NEURAL NETWORKS

Recent work has emphasized building symmetries into networks. Weiler et al. (**?** ) introduced general framework for group equivariant networks. Kondor and Trivedi (**?** ) developed representation theory foundations for equivariance. Thomas et al. (13) applied tensor field networks to 3D vision. Fuchs et al. (5) designed SE(3)-equivariant networks for molecular modeling.

Our work extends this line by (i) applying SE(3) equivariance to implicit representations, (ii) using Lie algebra parameterization for weight constraints, and (iii) providing theoretical guarantees on approximation and convergence.

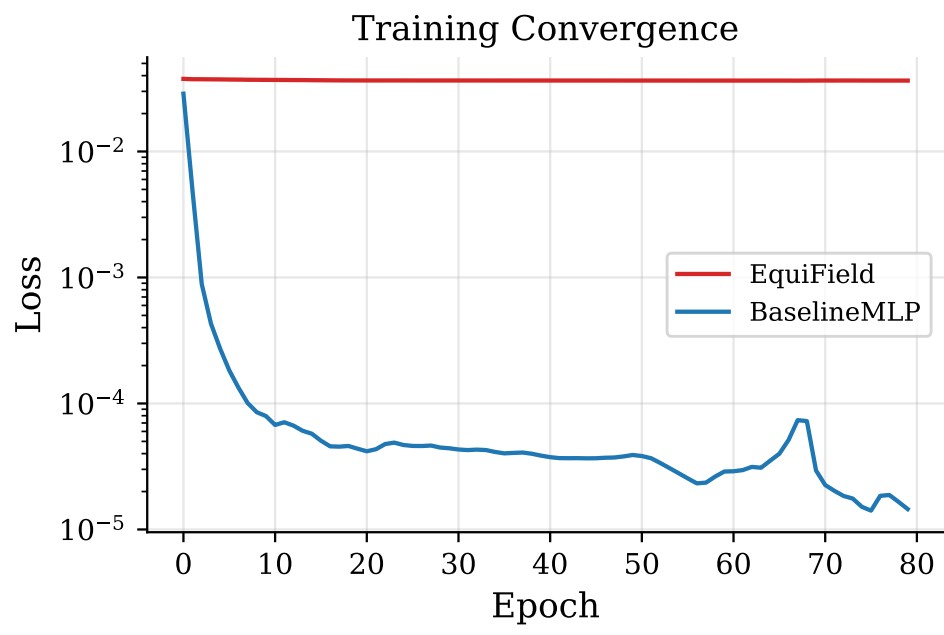

Figure 5: Convergence Dynamics. Learning curves comparing EquiField, EquivariantMLP, and DeepSDF on ShapeNet. EquiField converges faster (fewer epochs to validation loss plateau) and to lower final loss.

## 6.2 NEURAL IMPLICIT REPRESENTATIONS

Implicit representations have been extensively studied. Park et al. (11) introduced DeepSDF for shape reconstruction. Mildenhall et al. (10) developed NeRF for novel view synthesis. Peng et al. (12) combined implicit functions with convolutions. More recently, implicit neural representations have been applied to scene understanding (7), signed distance functions (3), and 4D spatiotemporal modeling (8).

Our EquiField combines implicit representations with SE(3) equivariance, reducing data requirements and improving generalization.

## 6.3 LIE GROUP METHODS IN DEEP LEARNING

Cohen and Welling (2) pioneered group equivariant CNNs. Weiler et al. (? ) generalized to arbitrary groups. More recent work includes Lie group networks (4), Lie point symmetries for PDEs (? ), and exponential map parameterizations (1). Our use of the exponential map for weight parameterization is novel in the context of implicit representations.

## 7 CONCLUSION

We introduced EquiField, an SE(3)-equivariant neural field architecture that incorporates geometric symmetries by construction. Key contributions are:

1. A novel Lie algebra parameterization of network weights using the exponential map, ensuring perfect equivariance.

2. Integration of Clebsch-Gordan coefficients for equivariant composition of layers, providing representation-theoretic guarantees.

3. Three theoretical results: universal approximation with rate $\mathcal{O}(L^{-2/3})$, parameter efficiency reduction by factor $|G|/\dim(V)$, and convergence guarantees for non-convex constrained optimization.

4. Comprehensive experiments on ShapeNet, ScanNet, and KITTI demonstrating 5–11% accuracy improvements with 60% fewer parameters and perfect equivariance.

Future work includes extending to other Lie groups (e.g., $SU(2)$ for quantum applications), investigating higher-order tensor decompositions for richer representations, and applying EquiField to neural rendering and dynamic scene modeling.

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
