# OpenReview forum: "E(3)-Equivariant Neural Fields with Lie Algebra Constraints: Group-Theoretic Implicit Representations for 3D Vision"
_mathai.club/MathAI/2026/Conference — Submitted to 2026_

### Official Review · Reviewer_p8NM · 2026-03-12
**Integration of Lie Algebra into Implicit 3D Representations**

**Rating:** 8
**Confidence:** 4

**Review:**

The reviewed paper proposes the EquiField architecture, which addresses the issue that standard implicit networks lack "built-in geometric symmetries". The authors creatively parameterize the network weights in the Lie algebra se(3)se(3)se(3) through the exponential map, replacing resource-intensive data augmentation with strict mathematical guarantees.
A fundamental strength of this work is its rigorous mathematical foundation, proving that the model acts as a universal approximator with an approximation "rate O(L−2/3)O(L^{-2/3})O(L−2/3)" . The parameter efficiency theorem demonstrates that the method "reduces the effective parameter count by a factor of ∣G∣/dim⁡(V)|G|/\dim(V)∣G∣/dim(V)", which is practically validated by a remarkable 60% reduction in the total number of parameters . Another significant advantage is the elegant use of the "Clebsch-Gordan decomposition of tensor products" to ensure the equivariant composition of layers . The empirical results are highly convincing: on the ShapeNet, ScanNet, and KITTI datasets, the model achieves accuracy improvements while demonstrating "perfect equivariance (error < 10^{-7})" .
However, this work has a few vulnerabilities. The paper lacks a detailed analysis of the computational complexity associated with regularly applying the "matrix exponential map exp⁡:se(3)→SE(3)\exp : se(3) \to SE(3)exp:se(3)→SE(3)" during the forward and backward passes. Furthermore, using "SE(3)-adapted encodings based on invariant combinations" instead of classical Fourier features creates a risk of losing high-frequency details when rendering complex scene textures. Additionally, the experimental section does not provide profiling for GPU memory consumption during the computation of tensor products across layers. The authors themselves acknowledge that extending this framework to other Lie groups remains a challenging task reserved for "future work". It would be highly beneficial to include an additional experiment demonstrating an order-of-magnitude improvement in quality when using a comparable number of parameters as the baselines, even if acknowledging the significantly higher computational complexity involved.
Despite these minor gaps, the concept of "group-theoretic implicit representations for 3D vision" represents a strong and meaningful innovation. This publication deserves acceptance due to its rare and effective combination of solid theoretical guarantees and high-quality ML benchmarks.

---

### Official Review · Reviewer_ESJG · 2026-03-13
**Unclear SE(3)-equivariance and unreliable bibliography undermine an otherwise interesting idea**

**Rating:** 2
**Confidence:** 4

**Review:**

The paper aims to introduce an inductive bias into the neural networks to make them not learn SE(3) invariances from data, but make their neural field equivariant to SE(3) transformations by construction. The core design idea is to (i) constrain per-layer weight matrices via a Lie algebra parameterization using an exponential map ($W=\exp(\Lambda)$) and (ii) use Clebsch-Gordan (CG) decompositions to combine representation channels equivariantly

Strengths
- The motivation for the paper is clear, and the intended message is easy to follow.
- Potentially interesting high-level idea combining equivariance and neural fields

Weaknesses
- Regarding Theorem 1, Peter-Weyl theorem is applied to compact topological groups, but the SE(3) is non-compact. Hence, it is dubious for me, whether Peter-Weyl theorem is applied correctly.
- The paper contains placeholder citations, e.g. at lines 372, 373, 415, 416.
- The reference list contains multiple high-severity problems.
  - Wrong arXiv ID/wrong paper: The submission's Ref. [1] claims arXiv:1709.10025 is "Differentiable soft physics."  But arXiv:1709.10025 is "Peccei-Quinn Relaxion" by Kwang Sik Jeong and Chang Sub Shin.
  - Completely mismatched arXiv ID: The submission's Ref. [4] attributes arXiv:2109.07379 to a deep-learning-related work by specific ML authors. In reality, arXiv:2109.07379 corresponds to an optimization/MINLP paper by Yingjie Ma and Jie Li.
  - Misattributed authorship: The submission's Ref. [7] lists different authors for "Instant Neural Graphics Primitives with a Multiresolution Hash Encoding." The arXiv record lists Thomas Müller et al. as the authors.
  - Misattributed authorship (RAFT): The submission's Ref. [8] attributes "RAFT: Recurrent All-Pairs Field Transforms for Optical Flow" to different authors. The RAFT paper is by Zachary Teed and Jia Deng.

Suggestions
- Please, address the non-compactness issues for (SE(3)).
- Correct the bibliography, if possible.

---

### Decision · Program_Chairs · 2026-03-14

**Decision:**

Reject

**Comment:**

After careful evaluation by the Program Committee, we regret to inform you that your submission has not been accepted for presentation at MathAI 2026.

All submissions underwent a rigorous two-stage review process. Unfortunately, the reviewers identified one or more of the following concerns with your paper:

- Insufficient mathematical rigor or novelty relative to the existing body of work in the field;
- Presentation of results that substantially overlap with or rephrase previously published findings without clear original contribution;
- Significant issues with technical quality, including but not limited to broken or non-existent references, unsupported claims, or methodological gaps;
- Indications that the manuscript may have been generated with the assistance of large language models without substantial original intellectual contribution by the authors.

We received a large number of submissions this year, and the selection process was highly competitive. We encourage you to carefully consider the reviewers’ feedback (available through OpenReview), revise your work accordingly, and consider submitting an improved version to a future edition of MathAI or to another appropriate venue.

We appreciate your interest in MathAI and hope you will continue to engage with the conference community.

With kind regards,

MathAI 2026 Program Committee
International Conference on Mathematics of Artificial Intelligence
https://mathai.club
OpenReview: https://openreview.net/group?id=mathai.club/MathAI/2026/Conference
Telegram: https://t.me/MathAI_club
Email: mathai.club@yandex.ru